# Molecular and Immune Phenotypic Modifications during Metastatic Dissemination in Lung Carcinogenesis

**DOI:** 10.3390/cancers14153626

**Published:** 2022-07-26

**Authors:** Drosos Tsavlis, Theodora Katopodi, Doxakis Anestakis, Savvas Petanidis, Charalampos Charalampidis, Evmorfia Chatzifotiou, Panagiotis Eskitzis, Paul Zarogoulidis, Konstantinos Porpodis

**Affiliations:** 1Department of Medicine, Laboratory of Experimental Physiology, Aristotle University of Thessaloniki, 54124 Thessaloniki, Greece; dr.tsavlis@yahoo.com; 2Department of Medicine, Laboratory of Medical Biology and Genetics, Aristotle University of Thessaloniki, 54124 Thessaloniki, Greece; katopodi@auth.gr; 3Department of Anatomy, Medical School, University of Cyprus, Nicosia 1678, Cyprus; anestaki@auth.gr (D.A.); ccharal@med.duth.gr (C.C.); 4Department of Pathology, Forensic Medical Service of Thessaloniki, 57008 Diavata, Greece; chatzifotiou.morfi@gmail.com; 5Department of Obstetrics, University of Western Macedonia, 50100 Kozani, Greece; peskitzis@gmail.com; 6Third Department of Surgery, “AHEPA” University Hospital, Aristotle University of Thessaloniki, 55236 Thessaloniki, Greece; pzarog@hotmail.com; 7Pulmonary Department-Oncology Unit, “G. Papanikolaou” General Hospital, Aristotle University of Thessaloniki, 57010 Thessaloniki, Greece; kporpodis@auth.gr

**Keywords:** disseminated tumor cells, immunosuppression, lung cancer, metastasis

## Abstract

**Simple Summary:**

Metastatic cancer is a multifaceted complex disease. It is mainly characterized by a strong invasive potential, metastasis, resistance to therapy, and poor clinical prognosis. Although the use of immune checkpoint inhibitors (ICI) has substantially improved cancer treatment and therapy, there are many significant challenges to be addressed. In this review, we provide an overview of the mechanisms used by metastatic or disseminating tumor cells (DTCs) in order to understand cancer progression to metastasis, and establish new strategies for novel therapeutic interventions.

**Abstract:**

The tumor microenvironment plays a key role in the progression of lung tumorigenesis, progression, and metastasis. Recent data reveal that disseminated tumor cells (DTCs) appear to play a key role in the development and progression of lung neoplasiaby driving immune system dysfunction and established immunosuppression, which is vital for evading the host immune response. As a consequence, in this review we will discuss the role and function of DTCs in immune cell signaling routes which trigger drug resistance and immunosuppression. We will also discuss the metabolic biology of DTCs, their dormancy, and their plasticity, which are critical for metastasis and drive lung tumor progression. Furthermore, we will consider the crosstalk between DTCs and myeloid cells in tumor-related immunosuppression. Specifically, we will investigate the molecular immune-related mechanisms in the tumor microenvironment that lead to decreased drug sensitivity and tumor relapse, along with strategies for reversing drug resistance and targeting immunosuppressive tumor networks. Deciphering these molecular mechanisms is essential for preclinical and clinical investigations in order to enhance therapeutic efficacy. Furthermore, a better understanding of these immune cell signaling pathways that drive immune surveillance, immune-driven inflammation, and tumor-related immunosuppression is necessary for future personalized therapeutic approaches.

## 1. Introduction

The metastatic spread of tumor cells is a key process for cancer progression. Accumulating evidence reveals that the spread of tumor cells can occur even at the early stages of carcinogenesis [1,2]. This process begins with the local invasion of primary tumor cells into surrounding healthy tissues: intravasation, extravasation formation of micrometastatic colonies, and the subsequent proliferation of microscopic colonies into metastatic lesions, a process termed as colonization (Figure 1). These disseminated tumor cells (DTCs) can migrate to distant organs and initiate tumor regrowth, triggering cancer recurrence and subsequent metastasis. Recent studies demonstrate that DTC-related tumor heterogeneity inside the tumor microenvironment (TME) plays a crucial role in tumor growth and expansion. The clinical significance of DTCs is proven by many studies, indicating their decisive and vital role in chemotherapy/immunotherapy resistance, tumor, recurrence, and metastasis [3,4]. In this review, we will investigate and analyze the various signaling pathways used by DTCs and their crosstalk with immune or stromal cells inside the TME, and we will also explore the development of new approaches and therapies that are specifically designed to target DTC-related metastasis.

## 2. Early Cancer Cell Dissemination

The mechanism of dissemination alters the phenotypic morphology of tumor cells in order to obtain traits that allow them to leave the primary tumor and spread (metastasize) to distant tissues or organs via blood circulation [5]. These circulating tumor cells (CTCs) can extravasate into secondary tissues, where they develop into DTCs. Active dissemination can occur through the transformation of epithelial cells into a more mesenchymal phenotype through an altered EMT process [6]. In many cases tumor cells disseminate in situ even before the detection of the primary tumor. This aspect of early dissemination, where DTCs develop in parallel with the primary tumor, is highly significant to understanding the unique molecular biology of metastatic colonization [2,7]. Recently, sustained lung inflammation was shown to convert disseminated, dormant cancer cells into aggressively growing metastases. This sustained inflammation induced the formation of neutrophil extracellular traps (NETs), and these were required for the awakening of dormant tumor cells [8]. In general, the TME is the critical regulator of cancer cell dissemination and cancer progression. This immunosuppressive and hypoxic TME can transform tumor cells in a prolonged quiescent state into active DTCs and in due course mediate metastatic outgrowth in distant organs or tumor recurrence [9,10].

### 2.1. Tumor Cell Dissemination

The peripheral blood contains numerous sources of tumor-derived material, like circulating tumor DNA (ctDNA), CTCs, and exosomes, which support tumorigenesis. Furthermore, the extracellular matrix (ECM) plays a critical role in the maintenance of the tumorigenic profile of DTCs [11,12]. These factors determine both the survival rate and adaptation of DTCs in metastatic sites. Apart from establishing metastatic hotspots, DTCs often come back to the primary tumor to accelerate their development [13]. This hypothesis of tumor self-feeding has broadened our knowledge of the pathogenesis of metastasis [14]. Furthermore, endothelial cells, which form the blood vessels that support the TME by providing nutrients and oxygen, also play critical roles in metastatic dissemination. Specifically, the tumoral activation of the TLR3-SLIT2 axis in endothelium drives metastasis. The endothelial SLIT2 protein and its receptor ROBO1 promoted the migration of cancer cells towards endothelial cells and intravasation. Deleting endothelial Slit2 suppressed metastatic dissemination in mouse models of breast and lung cancer. Conversely, the deletion of tumoral Slit2 enhanced metastatic progression [4]. Similarly, tumor-mesenchymal stem cells (T-MSCs) as heterogeneous stromal cells promote lung cancer metastasis by inducing the expression of genes associated with an aggressive phenotype in primary lung cancer cells [15]. Apart from that, immune cells can also promote the initial metastatic dissemination of carcinoma cells from primary tumors. For example, CD11b(+)/Ly6G(+) neutrophils enhance metastasis formation via the inhibition of the natural killer cell function, which leads to a significant increase in intraluminal survival along with the extravasation/dissemination of tumor cells through the secretion of IL1β and matrix metalloproteinases [16].

### 2.2. Epithelial-Mesenchymal Plasticity

Both the epithelial to mesenchymal transition (EMT) and mesenchymal to epithelial transition (MET) are developmental programs that are activated during embryogenesis and tissue repair [17]. During carcinogenesis, these programs are being hijacked by tumor cells in order to increase their resistance to chemotherapy, acquire the cancer stem cell (CSC) phenotype, and amplify their metastatic dissemination potential [18,19] (Figure 2). DTCs can manipulate EMT signaling in order to promote their dissociation from the primary tumor and disseminate into blood circulation [20]. Because EMT/MET are highly dynamic and plastic, DTC employ several EMT-inducing transcription factors such as Twist, Snail, Slug, and Zeb1, which activate these mechanisms and prompt metastatic colonization [21]. The activation of EMT can also occur via epidermal growth factor (EGF), fibroblast growth factor (FGF), hepatocyte growth factor (HGF), insulin growth factor (IGF), and PDGF. In many cases, DTCs cells transit through a series of EMT/MET states also named partial-EMT, where tumor cells possess both epithelial and newly acquired mesenchymal characteristics and display a high degree of phenotypic plasticity [22,23].

## 3. Integrin Signaling

Integrins are transmembrane proteins that mediate cell adhesion, regulate the cell cycle, and organize the intracellular cytoskeleton [24]. Accumulating evidence reveals that tumor cells alter integrin signaling inside the TME, thus contributing to immunosuppression and immunoresistance due to their ability to control immune cell adhesion to endothelial cell layers followed by their trans-migration [25]. They are also involved in tumor expansion and progression by mediating angiogenesis, lymphangiogenesis, desmoplasia, and inflammation [26,27]. Tumor cell expression of the integrins αvβ3, αvβ5, a5β1, a6β4, a4β1, and αvβ6 is often associated with increased chemoresistance, disease progression, and poor survival [28,29].

### Metastatic Colonization

Recent data reveal that in the early stages of the metastatic cascade, bidirectional communication between malignant cells and the TME is essential for maintaining tissue homeostasis and tumor expansion. Metastatic tumorigenesis is a heterogeneous disease progressing in a multistep process involving aberrant mutated tumor cells, altered phenotype immune cells, and highly polarized stromal cells. As a whole, metastatic colonization is a very ineffective mechanism, where only 0.01% of tumor cells survive to form micro-metastases. During this early phase of metastasis, cytoskeleton rearrangements within tumor cells in cooperation with aberrant differentiation signaling of Notch, WNT, and Hedgehog pathways guide tumor cell invasion. This metastatic cascade occurs via single tumor cells or cell clusters. Recently, the ability of CTCs to form clusters has been linked to increased metastatic potential. In particular, CTC clustering shapes DNA methylation to enable metastasis seeding and the promotion of stemness [30]. These CTC clusters can derive from multicellular groupings of primary tumor cells held together with intercellular adhesion, and they greatly contribute to the metastatic spread of cancer [31]. Most CTC clusters are heterogeneous and adopt multiple ways to enhance their metastatic potential, including homotypic clustering and heterotypic interactions with immune and stromal cells [32,33].

## 4. Dormancy Niche

In a plethora of tumors, dormant DTCs can reside in distinctive compartments inside the TME which sustain their proliferation and survival from immune surveillance. Dormant or quiescent DTCs may lay dormant for long periods and are characterized by the reversible growth arrest in the G0-G1 phase of the cell cycle. This tumor dormancy is characterized by a balance between tumor cell proliferation and apoptosis [34]. Endowing quiescent DTCs with stem-like characteristics leads to their activation and metastatic outgrowth. For that reason, dormant DTCs have developed several in-house mechanisms for their initial and long-term survival. For instance, autophagy is a critical mechanism for the survival of dormant breast DTCs. The pharmacologic or genetic inhibition of autophagy in dormant BC cells leads to a significant decrease in cell survival and metastatic burden in mouse and human 3D in vitro and in vivo preclinical models of dormancy [35]. Furthermore, in BC patients NRF2 signaling induces a transcriptional metabolic reprogramming to re-establish redox homeostasis and upregulate de novo nucleotide synthesis. NRF2 is activated during dormancy in recurrent tumors in animal models and in patients with poor prognosis. The constitutive activation of NRF2 accelerates recurrence in recurrent tumors [36]. In many cases, dormant DTCs reside inside the perivascular niche (PN) in the brain, lung, and bone marrow. Precise factors within the PN have been shown to actively promote dormancy. For example, thrombospondin-1, produced from surrounding endothelial cells, was able to confine DTCs in a quiescent state, whereas TGF-β1 and periostin triggered tumor-promoting micrometastatic outgrowth [37]. In conjunction with this mechanism, T cells can also promote the dormancy mechanism. A discrete population of CD39+PD-1+CD8+ T cells mediates metastatic dormancy in breast cancer. These distinct cells are only found in primary tumors and in dormant metastasis, but they are hardly found in aggressively metastasizing tumors [38].

### 4.1. CSC Programming

DTCs are mainly characterized by their EMT characteristics. This EMT transition can mediate both the dissemination and possible acquisition of CSC traits and phenotype by tumor cells. Recently, it was shown that the bone microenvironment facilitates breast and prostate cancer cells further metastasizing and establishing multi-organ secondary metastases. This metastasis-promoting effect is driven by epigenetic reprogramming that confers stem cell-like properties to cancer cells [39]. In addition, in pre-malignant lesions, CCL2 produced by cancer cells and myeloid cells attracts CD206^+^/Tie2^+^ macrophages and induces Wnt-1 upregulation that, in response, downregulates E-cadherin junctions in the HER2^+^ early cancer cells. The formation of macrophage compartments in the pre-malignant lesions was detected, and they can operate as portals for intravasation [40]. Recent data reveal that dormant DTCs are CSCs or have acquired stem-like characteristics. For instance, self-renewal, differentiation, and chemoresistance are also features of dormant DTCs that give rise to metastatic disease [41]. A recent hypothesis states that DTCs can reside in the CSC state, waiting for the optimal conditions inside the TME to trigger metastasis. For example, colorectal cancer metastases are seeded by Lgr5^−^ negative cells, which display an intrinsic capacity to become CSCs in a niche-independent manner and can restore epithelial hierarchies in metastatic tumors [42]. Moreover, tumor cells display a phenotypic equilibrium between the stem-like and differentiation states during tumor homeostasis. Dormant DTCs can also reprogram the TME to initiate the metastatic signaling. Equally, visceral adipose stromal cells (V-ASCs) can reprogram the colorectal cancer stem cell metastatic machinery by recruiting adipose stem cells within the tumor mass, promoting vasculogenesis and the onset of metastatic dissemination by the activation of STAT3/ZEB2 expression and effectively reprogramming CRC cells into a highly metastatic phenotype [43].

### 4.2. Tumor Microenvironment

Primary tumors are highly dependent on their residual stromal microenvironment. For DTCs residing at distant organs, their dormant quiescent state is mainly regulated by the local microenvironment. The transition of tumor cells from a dormant state to an activated outgrowth state may be provoked by changes in their local microenvironment. The increased expression of specific factors such as TGF-β/TGF-β2, CXCL12, bone morphogenic protein 7 (BMP7), leukemia inhibitory factor (LIF), and Wnt signaling can break dormancy and promote tumor cell proliferation [44]. Dormancy programs can be initiated through the p38MAPK pathway together with decreased integrin-mediated mitogenic signaling, which leads to cell cycle G0/G1 arrest and associated quiescence. Likewise, BMP7 secreted from bone stromal cells induces senescence in prostate CSCs by activating p38 mitogen-activated protein kinase and increasing the expression of the cell cycle inhibitor, p21, and the metastasis suppressor gene, NDRG1, which correlates with recurrence and bone metastasis in prostate cancer patients [45]. In parallel, TGF-β2 dictates disseminated tumor cell fate in target organs through TGF-β-RIII and p38α/β signaling, revealing a ‘seed and soil’ mechanism where TGF-β2/TGF-β-RIII signaling regulates DTC dormancy and defines restrictive (BM) and permissive (lung) microenvironments for HNSCC metastasis [46]. The reactivation of dormant DTCS has also been reported in a recent study where osteoblast-secreted factors mediated the dormancy of metastatic prostate cancer in the bone via the activation of the TGFβRIII-p38MAPK-pS249/T252RB pathway [47].

## 5. Immune Suppression

Accumulating evidence shows that the interactions of DTCs with the stromal cells of the TME prompt immune system inhibition and immunosuppression (Figure 3). These stromal cells that support the immunosuppressive metastasis machinery are primarily regulatory T cells (Tregs), myeloid-derived suppressor cells (MDSCs), tumor-associated macrophages (TAMs), and dendritic cells (DCs). In particular, DCs are responsible for coordinating antitumor immunity. In many tumors, alternatively activated DCs can trigger antigen-specific tolerance via CD4+ T cell differentiation into Tregs and prompt CD8+ T exhaustion or anergy [48]. This manipulation of DCs by tumor cells, especially DTCs, can result in the perseverance of ‘immune-edited’ tumors that escape immunosurveillance and guide cancer relapse [49]. This tumor-associated immune dysfunction protects tumor cells from T cell elimination and may support tumor expansion and dissemination via circulation to distant organs [50]. Equally, MDSCs, which are defined as a heterogeneous population of immature myeloid cells, play a significant role in the immunosuppression and chemoresistance of both primary and metastatic tumors. In a plethora of tumors, MDSCs accumulate in the TME and establish early immune suppression by suppressing cytotoxic T cell activity via Treg/HIF-1a induction and through the production of the reactive oxygen species (ROS), arginase 1 (Arg-1) and inducible nitric oxide synthase (iNOS) [51]. The mobilization of MDSCs’ differentiation and migration to the TME core is orchestrated by immunosuppressive cytokines such as GM-CSF, VEGF, IL-6, IL-10, TGF-β, and other factors [52].

### Angiogenesis

A significant factor in tumor cell dissemination is the formation of new blood vessels via angiogenesis in order to support tumor growth and expansion. VEGF is the main coordinator of new blood vessel formation from pre-existing endothelial cells on blood vessels [53]. Tumor cells and DTCs inside the hypoxic TME secrete high levels of pro-angiogenic factors such as basic fibroblast growth factor (bFGF), VEGF, PDGF-B, and TGF-β, which are key factors in juvenile vascular network formation in tumors [54]. Recently, the growth, metastasis, and angiogenesis of small cell lung cancer (SCLC) by profilin 2 (PFN2) was reported, which triggered vascular formation and angiogenesis via tumor-derived exosomes [55]. In many cases, the inflammatory TME contributes to metastasis by recruiting blood and lymph vessels, such as the S1PR1 on tumor-associated macrophages (TAMs), which triggers lymphangiogenesis, tumor angiogenesis, and metastatic spread via NLRP3/IL-1β in pulmonary cancer [56]. Likewise, in metastatic TAMs, the loss of caveolin-1 drives lung metastatic growth through increased angiogenesis via restraining VEGF-A/VEGFR1 signaling and its downstream effectors, matrix metallopeptidase 9 (MMP9) and colony-stimulating factor 1 (CSF1) [57]. In a similar manner, DUSP1/MKP1, a specific phosphatase that regulates MAPKs activity, and RAGE (receptor for advanced glycation end-product) were associated with the promotion of angiogenesis, invasion, and metastasis in NSCLC patients [58,59]. Several other transcription factors and receptors are also involved in angiogenesis signaling, such as ELF5 and Cysteinyl leukotriene 2 receptor (CysLT2R), which trigger endothelial permeability, tumor angiogenesis, and lung metastasis by leukocyte infiltration and increased blood vessel permeability [60,61].

## 6. Therapeutic Clinical Applications

The advances of modern tumor immunotherapy approaches have revolutionized the field of cancer research. Several small clinical trials targeting DTCs post-treatment are really promising, and second-line chemotherapy regimens such as bisphosphonates (zoledronate, ibandronate) show promising and encouraging results. Furthermore, the advancement of single-cell analysis techniques, mainly single-cell RNA sequencing and immune signaling pathway profiling, is vital for the phenotypic analysis of tumor cell heterogeneity. A better understanding of the crosstalk between DTCs and the tumor immune microenvironment (Table 1) may also hold the key for overcoming chemoresistant tumor relapse and metastasis [62]. In addition, several clinical trials using single agents and combination therapies focus on the inhibition of developmental signaling pathways that are crucial for stem and progenitor cell homeostasis, such as the Notch, WNT, Hedgehog, and Hippo signaling cascades [41]. A deeper understanding of the factors that induce dormancy and tumor cell dissemination is crucial for inhibiting factors that trigger the activation of dormant DTCs. In particular, the isolation and identification of DTCs or CTCs by using specific cell surface markers, such as CD133, CD44, EpCAM, or aldehyde dehydrogenase (ALDH), remains a true challenge in preventing metastatic spread and tumor relapse [63]. Additionally, the use of IGF-1R along with KDM or HDAC inhibitors has been shown to effectively eradicate dormant cells [64]. Recent therapeutic strategies based on leukapheresis, and autologous stem cell transplantation can also display encouraging results. Overall, these emerging concepts can provide a better understanding of metastatic colonization, an essential step to improving metastatic tumor therapy.

## 7. Conclusions

Metastatic dissemination is a complex multistep process. An ideal systemic therapy should target the molecular immune-related mechanisms in the tumor microenvironment that lead to decreased drug sensitivity and tumor relapse, and also reverse drug resistance and tumor-induced immunosuppression. Therefore developing specific therapeutic strategies in order to inhibit metastatic colonization will be a promising direction in lung cancer treatment.and will radically enhance therapeutic efficacy

## Figures and Tables

**Figure 1 cancers-14-03626-f001:**
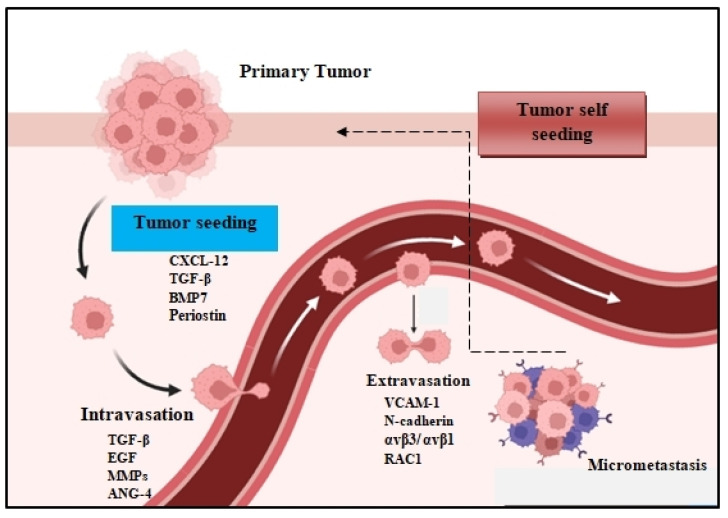
The metastatic colonization cascade. The various steps of metastasis are depicted: intravasation, circulation, extravasation, and colonization of distant organs. In many tumors, cancer cells can return to the primary tumor (tumor self seeding) and accelerate its expansion.

**Figure 2 cancers-14-03626-f002:**
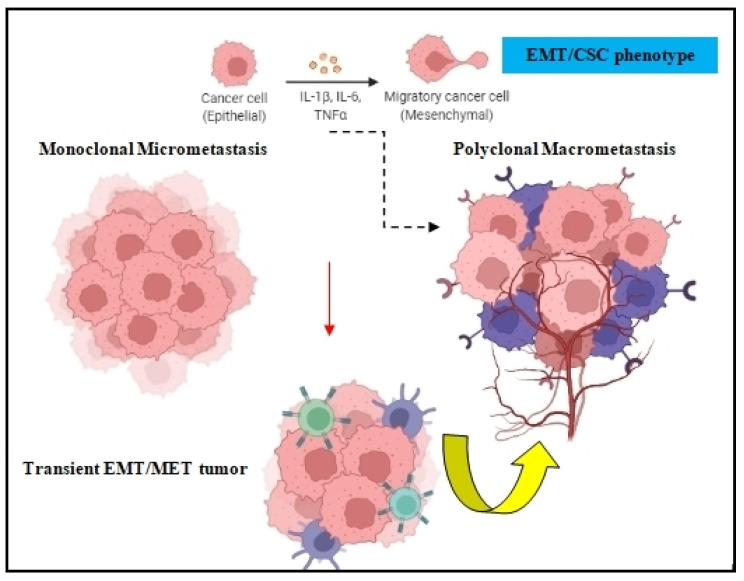
EMT in tumor cell dissemination. The EMT/CSC phenotype of DTCs can prompt monoclonal micrometastasis seeding to develop polyclonal macrometastasis, depending on the clonal and tumor-stromal cells interactions in the target organ.

**Figure 3 cancers-14-03626-f003:**
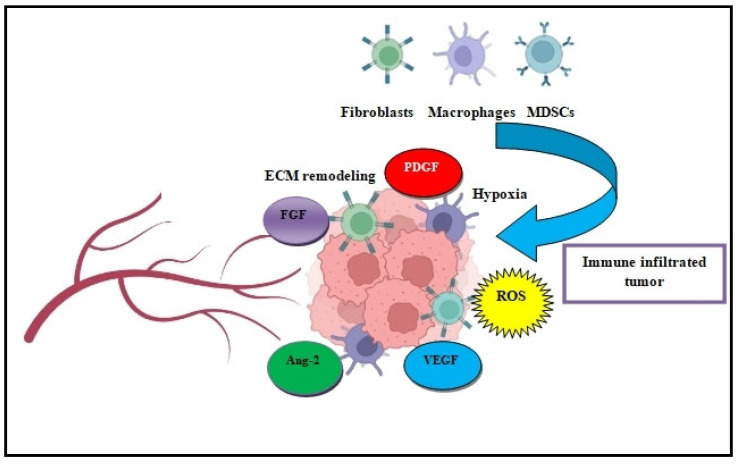
Immune cell infiltration is vital for DTC`s activity. Crosstalk of DTCs with immune cells in the TME prompts ECM remodeling, hypoxia, and immunosuppression. Subsequently, the secretion of angiogenesis-related factors such as Ang-2 FGF, PDGF and VEGF triggers further tumor growth and expansion.

**Table 1 cancers-14-03626-t001:** Overview of the role and function of disseminated tumor cells in immune cell signaling pathways, drug resistance, and immunosuppression.

DTCs	Function	References
DTCs	Inflammation	[9]
DTCs	Metastasis	[4,17,18]
DTCs	EMT	[23,24,25]
DTCs	Chemoresistance	[30,31,43]
DTCs	Stemness	[32,33,34,35]
DTCs	Autophagy	[37]
DTCs	Metabolic reprogramming	[38]
DTCs	Dormancy	[39,40]
DTCs	CSC programming	[41,42,43,44,45,47]
DTCs	Immune suppression	[51,52,53,54,55]
DTCs	Angiogenesis	[57,58,59,60,61,62,63]

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
