# Peer review of "Molecular and Immune Phenotypic Modifications during Metastatic Dissemination in Lung Carcinogenesis"

_cancers, 2022, doi:10.3390/cancers14153626_

Round 1
Reviewer 1 Report
In this manuscript "Molecular and immune phenotypic modifications during meta-2 static dissemination in lung carcinogenesis" Drosos and colleagues focused on the roles of the immune system in the progression of metastasis. It is the interest of the topic and the quality of the review. However, I have some major concerns about this manuscript.
1. The hierarchy of this article seems to need to be more clear. Headings like 1, 2, 3… may not be enough (a bit confusing), subheadings 2.1, 2.2...are needed.
2. CSC first appears without a full name.
3. DTC and dormancy are closely related, which may require a more detailed discussion.
4. There seems to be too little immune-related content, which does not match the highlighted focus of the title. More detailed?
5. Sections 7 and 9 cross over and need to be rewritten more clearly.
6. The last part is too simple and should be described in more detail. For the treatment of dormancy, this article may be useful (10.1016/j.tips.2018.12.004).
Author Response
Response to reviewer comments:
We thank the reviewers for their interest in our manuscript, positive and helpful
suggestions. Provided below is a point-by-point response describing our attempts to address, and when possible incorporate, all of their requested revisions in our manuscript.
Reviewer Comments
Comment: The hierarchy of this article seems to need to be more clear. Headings like 1, 2, 3… may not be enough (a bit confusing), subheadings 2.1, 2.2...are needed.
Response: We thank the Reviewer for his useful comment. Appropriate subheadings were inserted were in the manuscript according to reviewer request.
Comment: CSC first appears without a full name
Response: We appreciate the comment made by the Reviewer. The full name of the abbreviation of CSC was added in the manuscript.
Comment: DTC and dormancy are closely related, which may require a more detailed discussion
Response: We are grateful to Reviewer for his valuable comment. DTCs and
dormancy were discussed in more detailed in the manuscript.
Comment: There seems to be too little immune-related content, which does not match the highlighted focus of the title. More detailed?
Response: We thank the reviewer for his useful comment. However, in the
manuscript, the role of DTCs in almost all immune phenotypic signaling pathways (methylation, inflammation, metabolic reprogramming, immunosuppression, angiogenesis, stemness, chemoresistance, etc) was discussed broadly and in detail. Since there is a plethora of information about these pathways we selected appropriate content to fit the theme of the manuscript. Nevertheless, additional information regarding immune-related content was added through the manuscript according to Reviewer`s request.
Comment: Sections 7 and 9 cross over and need to be rewritten more clearly.
Response: We appreciate the comment made by the Reviewer. For that reason, both Sections 7 and 9 were rewritten more clearly according to Reviewers suggestion.
Comment: The last part is too simple and should be described in more detail. For the treatment of dormancy, this article may be useful (10.1016/j.tips.2018.12.004).
Response: We thank the Reviewer for his useful comment. The last section of the manuscript was rewritten in more detail and appropriate information was added. For dormancy treatment, the above article was cited according to Reviewer`s request.

Reviewer 2 Report
The authors in this review, provide an overview of the mechanisms used by metastatic or disseminated tumor cells to understand cancer progression to metastasis and establish new strategies for novel therapeutic interventions.
The manuscript is interesting and well supported by figures of the object of the study. However, I believe that it needs to add a table representing the role and function of disseminated tumor cells in immune cell signaling pathways, drug resistance, and immunosuppression.
Thank you for allowing me to review this interesting manuscript.
Author Response
Response to reviewer comments:
We thank the reviewers for their interest in our manuscript, positive and helpful suggestions. Provided below is a point-by-point response describing our attempts to address, and when possible incorporate, all of their requested revisions in our manuscript.
Reviewer Comments
Comment: The authors in this review provide an overview of the mechanisms used by metastatic or disseminated tumor cells to understand cancer progression to metastasis and establish new strategies for novel therapeutic interventions. The manuscript is interesting and well supported by figures of the object of the study. However, I believe that it needs to add a table representing the role and function of disseminated tumor cells in immune cell signaling pathways, drug resistance, and immunosuppression.
Response: We thank the Reviewer for his useful comment. A table describing the role and function of disseminated tumor cells in immune cell signaling pathways, drug resistance, and immunosuppression was added, according to reviewer`s request.

Round 2
Reviewer 1 Report
The revised manuscript has addressed the concerns mentioned earlier.